# Targeting Capabilities of Native and Bioengineered Extracellular Vesicles for Drug Delivery

**DOI:** 10.3390/bioengineering9100496

**Published:** 2022-09-22

**Authors:** Liubov Frolova, Isaac T. S. Li

**Affiliations:** Department of Chemistry, The University of British Columbia, Kelowna, BC V1V 1V7, Canada

**Keywords:** extracellular vesicle, exosome, drug delivery, endocytosis, drug targeting

## Abstract

Extracellular vesicles (EVs) are highly promising as drug delivery vehicles due to their nanoscale size, stability and biocompatibility. EVs possess natural targeting abilities and are known to traverse long distances to reach their target cells. This long-range organotropism and the ability to penetrate hard-to-reach tissues, including the brain, have sparked interest in using EVs for the targeted delivery of pharmaceuticals. In addition, EVs can be readily harvested from an individual’s biofluids, making them especially suitable for personalized medicine applications. However, the targeting abilities of unmodified EVs have proven to be insufficient for clinical applications. Multiple attempts have been made to bioengineer EVs to fine-tune their on-target binding. Here, we summarize the current state of knowledge on the natural targeting abilities of native EVs. We also critically discuss the strategies to functionalize EV surfaces for superior long-distance targeting of specific tissues and cells. Finally, we review the challenges in achieving specific on-target binding of EV nanocarriers.

## 1. Introduction

Extracellular vesicles (EVs) are membranous nanosized particles produced by nearly all cell types, including eukaryotic and prokaryotic cells, and they carry their parent cell’s cytosolic components in their lumen, including RNA and various proteins. EVs can be broadly classified into three types by their biogenesis pathway: exosomes, microvesicles (ectosomes) and apoptotic bodies. Exosomes have attracted the most attention in the drug delivery field due to their unique composition and their ability to deliver messenger proteins and genetic components to distant sites. Exosomes are 30 to 150 nm in diameter and their generation starts with receptor-mediated endocytosis at the cell’s plasma membrane. The endocytosed contents are directed towards the endosome which later matures into a multivesicular body (MVB) filled with intraluminal vesicles. Upon fusion of the MVB with the plasma membrane, these intraluminal vesicles are released into the extracellular space as exosomes. Microvesicles directly pinch off the cell’s plasma membrane. While they are less commonly researched for their drug delivery capabilities, microvesicles from specific cell types, especially red blood cells, are promising delivery tools due to their scalability. Apoptotic bodies are released by dying cells and are outside the scope of this review.

EVs’ natural capacity to penetrate hard-to-reach tissues is only matched by a limited number of highly advanced synthetic nanocarriers. For example, EVs have a unique ability to cross the blood-brain barrier (BBB)—a major challenge faced by traditional therapeutics. The mechanism behind this is poorly understood, but exosomes are hypothesized to be transported to the brain via vesicular transport, known as transcytosis [1]. EVs also show promise in hearing loss treatment by delivering drugs to the cochlear sensory hair cells, which are particularly inaccessible because the blood-labyrinth barrier and multiple tissue barriers hinder the access of systemically and locally administered drugs [2,3]. EVs have the potential to effectively penetrate tissues because they are better suited to the stress-relaxation environment of the extracellular matrix and diffuse more readily through its dense mesh compared to liposomes of similar size [4].

Further, EVs are believed to be more likely to have extended circulation times compared to synthetic nanoparticles due to their stability and ability to evade the immune system. The phospholipid surface of EVs is negatively charged, potentially making it easier for them to evade macrophages and cells of the mononuclear phagocyte system (MPS), which are also negatively charged [5,6]. Additionally, some surface proteins enriched in exosomes, such as CD47 [7], help with MPS evasion [8]. Exosomes derived from antigen-presenting cells are particularly good at evading the immune system and avoiding lysis by the complement system due to the enrichment of CD55 and CD59 markers on their surface [9].

Unlike synthetic carriers, EVs from certain cell types possess intrinsic therapeutic qualities that may supplement the effects of the incorporated drugs. This is particularly true for stem cell-derived exosomes. For example, EVs derived from mesenchymal stem cells (MSC) have anti-inflammatory and protective properties similar to their parent cells. Multiple studies have shown the regenerative potential of such EVs [10,11,12,13]. Exosomes from tendon stem cells have been shown to enhance tendon repair after injury [14], while the cardiac stem and progenitor cell-derived exosomes heal the heart muscle after an infarction [15,16].

Finally, when stored at −80 °C, EVs remain stable and retain clinical usability even after five months of storage [17,18]. This property of EVs is of high practical significance if they are to be commercialized as drug delivery vectors.

EVs have shown enough promise as drug delivery vectors to become the subject of several high-value pharmaceutical deals. In particular, the unique ability of EVs to cross the blood-brain barrier has attracted over CAD 1 billion of investments from Eli Lilly into central nervous system (CNS)-targeting exosomes, which could be loaded with RNA interference (RNAi) and antisense oligonucleotide (ASO) therapeutics. The addition of guiding ligands will enhance CNS targeting. While there is a definite focus on exosomal nucleic acid delivery (seven related big pharma deals in 2020 alone), adult stem cell companies also show considerable interest in EVs [19,20]. It is believed that exosomes carry the same paracrine factors for tissue regeneration as their parent stem cells, but are easier to scale up and modify for specific purposes [20]. It is hoped that EVs can be used to make therapeutics which will be uniquely biocompatible and suited to the patient’s disease stage. This can be accomplished by isolating EVs from the patients’ primary cells as well as from body fluids, e.g., blood, semen, saliva, tears, etc., and loading them with the appropriate drug dose.

Clinical trials are continuously initiated to assess EVs’ suitability for various therapeutic applications. Dendritic-cell (DC)- and MSC-derived EVs are the most common types of EVs used in clinical trials. A phase II/III clinical trial showed that the use of EVs derived from MSCs improved kidney function in patients with chronic kidney disease and did not produce significant side effects [21]. A phase II trial tested DC-derived exosomes loaded with tumour antigens as vaccines against unresectable non-small cell lung cancer (ClinicalTrials.gov Identifier: NCT01159288) [22]. Although no specific T cell response against cancer cells was observed, the vaccine induced natural killer cell activation in patients. Unfortunately, the trial was terminated because the primary endpoint of observing at least 50% of patients with progression-free survival at 4 months after stopping chemotherapy was not reached.

The clinical use of EVs as delivery vehicles for various therapeutics is currently being tested. For example, a phase II trial sought to treat hard-to-manage malignant ascites and pleural effusion in cancer patients by administering various chemotherapeutic drugs packaged into cancer EVs (ClinicalTrials.gov Identifier: NCT01854866). The results of this trial are, however, unknown. Another ongoing study will investigate the ability of plant-derived exosomes packaged with curcumin to deliver their contents to normal colon tissue and colon tumours (ClinicalTrials.gov Identifier: NCT01294072).

Recently, there has been a surge of interest in using EVs to treat COVID-19. Currently, eight ongoing and two completed clinical trials are registered on ClinicalTrials.gov, which use exosomes as a treatment option for COVID-19. Two ongoing studies (phase I and phase II) are investigating the safety and efficacy of CD24-exosomes, which can potentially suppress the cytokine storm in patients with moderate or severe COVID-19 (ClinicalTrials.gov Identifiers: NCT04747574 and NCT04902183). Another two trials (phase I and II) are looking into the safety and efficacy of bone marrow MSC-derived exosomes in severely ill patients hospitalized with COVID-19 (ClinicalTrials.gov Identifiers: NCT04602442 and NCT04276987).

Given the amount of dedicated effort and investment into EV-based therapeutics, it is vital to ensure that they are targeted to the tissues of interest to maximize their medicinal action and minimize off-target effects. This review will focus on EVs’ natural targeting abilities and the bioengineering strategies to target specific cells. We will also discuss the challenges and future perspectives of EV-based therapeutics.

## 2. Unmodified EVs Have Limited Intrinsic Targeting Abilities

EVs are nature’s delivery vehicles, as their surface is decorated with numerous molecules, such as tetraspanins, integrins and cell-specific proteins, which serve as native targeting moieties [23]. These molecules are instrumental to EVs’ role in intercellular communication as they ensure that they reach their target tissues, successfully dock onto the specific cells and offload their cargo. EVs exhibit organotropic behaviour in local cell-cell communication. However, they have also been shown to facilitate cargo delivery to distant sites. Such long-range organotropism is a highly sought-after property in drug delivery vehicle design. Therefore, EVs’ intrinsic targeting abilities have become the subject of particular interest.

Long-range organotropism of cancer EVs is well-documented. Cancer EVs are known to travel far from their original tumour sites and alter the healthy tissues, thus preparing the ‘soil’ for metastatic ‘seeds’ to be planted [24,25,26,27]. This has prompted researchers to study the natural abilities of cancer EVs to target certain cells and tissues. Cancer exosomes have been shown to preferentially fuse with their parent cell types compared to other cells both in vitro and in vivo when injected systemically in tumour-bearing mice [28]. Further, prostate cancer EVs were shown to efficiently deliver a therapeutic agent to their parent cells [29]. Exosomes from brain cells (endothelial or tumour) crossed the blood-brain barrier and delivered their therapeutic cargo to brain tumours in a zebrafish model [30]. The apparent propensity of cancer EVs to migrate and fuse with their parent cells is most likely due to the lipid and protein composition (especially surface receptors and extracellular matrix-binding proteins) of EVs, which uniquely resembles that of their parent cells. Cancer exosomes may also preferentially target certain healthy organ tissues, such as Kupffer cells in the liver and fibroblasts and epithelial cells in the lung, depending on their cancer cell line of origin [24]. The expression pattern of integrins on exosomes’ surfaces is deemed to determine their adhesion to specific cell types [24]. The differences in the tetraspanin complexes incorporated into exosomal membranes also impact target cell selection both in vitro and in vivo [31,32]. Further, when injected directly into the brain, exosomes from neuroblastoma cells bind to amyloid (plaque deposits) and shuttle them to brain microglial immune cells for removal [33]. That said, cancer exosomes still fuse more efficiently with cancer cells than other cell types, with acidic tumour conditions being one of the key factors in directing exosomal traffic [34]. Strikingly, compared to similarly sized liposomes, cancer exosomes show a more than 10-fold greater association with cancer cells [35].

The targeting abilities of EVs originating from non-cancerous cells have also been explored. For example, EVs derived from immune cells appear to preferentially target immune cells. For instance, exosomes originating from T cells target antigen-presenting cells for unidirectional microRNA transfer [36]. This exosome-mediated communication process is important for antigen recognition in an immune response. An excellent review on the role of EVs in immune response and the targeting of drug-loaded EVs for cancer immunotherapy has recently been published by Ruan et al. [37]. Further, unmodified EVs from immortalized human embryonic kidney cells have been shown to accumulate in tumour tissues in vivo [38]. This effect is consistent with the findings for similarly sized nanoparticles, such as liposomes, which also tend to accumulate in tumours. The vasculature of tumours is abnormal with leaky vessels, which enables extravasation (permeation from blood vessels into surrounding tissue) of nanoparticles [39]. This enhanced permeation and retention effect can be further optimized for more specific targeting of tumours by choosing different cell lines to obtain EVs.

A study comparing the in vivo distribution of EVs originating from cancer cells vs. non-cancerous cells have shown that cellular origin plays a role in the fate of exogenously administered EVs. Thus, in a mouse model, melanoma-derived EVs were shown to target the gastrointestinal tract more than those from non-cancerous cells [38]. On the other hand, EVs from bone marrow dendritic cells accumulated in the spleen to a greater extent than the EVs of melanoma and muscle cell origin [38]. Curiously, the species origin of the EVs (human, rat or mouse) did not affect the in vivo distribution [38].

In addition to choosing the appropriate parent cell type, altering the route of administration can be used to increase the targeting of EVs towards the tissue of interest. In a mouse model, intravenous (iv) injection resulted in the highest accumulation of EVs in the liver compared to intraperitoneal (ip) and subcutaneous (sc) injection (~60% vs. ~30%) [38]. Ip and sc injection routes showed lower EV accumulation in the spleen but increased accumulation in the pancreas and gastrointestinal tract compared to iv injection [38].

Despite the evidence discussed above, the ability of unmodified EVs to specifically target cells and tissues in vivo remains controversial. A study by Jung et al. demonstrated that exosomes derived from MDA-MB-231 human breast cancer cells only targeted hypoxic cancer cells in vitro, but failed to do so in vivo [40]. Further, multiple in vivo studies have shown that, upon systemic administration, EVs accumulate non-specifically in the liver and spleen, with a proportion directed towards kidneys, lungs and gastrointestinal tract [38,41,42,43]. This distribution pattern is similar to other systemically administered nanoparticles, such as liposomes. Smyth et al. showed that tumour-derived exosomes did not accumulate in tumour tissues unless directly injected into the tumour site [42]. Thus, although unmodified EVs display some specificity to certain cell types, they do not demonstrate sufficiently precise targeting for clinical applications [44]. Instead, the findings about natural cellular and tissue tropism of EVs should be used to design safe, targeted drug delivery vehicles. Murphy et al. have written an excellent review comparing the intrinsic targeting abilities of EVs with their engineered counterparts [45].

## 3. Engineering EVs for Targeted Drug Delivery

EVs can be decorated with surface molecules to enhance their targeting abilities. This can be accomplished by directly attaching targeting moieties to the EV surface or modifying EV-producing cells (See Figure 1). A very detailed review on this topic covering the targeting of EVs via both the direct chemical modification of EVs’ surface and genetic modification of their parent cells has been published by Liang et al. [46].

### 3.1. Targeting EVs by Modifying Parent Cells

Modifying parent cells is a common method to obtain EVs with particular targeting properties and is superior to direct modification of exosomal surface in terms of the stability of the targeting moiety. In this method, a gene encoding the targeting proteins is inserted into donor cells, and the cells then release EVs carrying those proteins via the natural biogenesis pathways. If certain targeting moieties are not naturally found on EV membranes, cells can be made to express the desired moiety fused to an EV membrane component.

Lysosomal-associated membrane protein 2b (Lamp2b) is one of the most commonly used exosomal pedestals to attach guiding moieties [47,48,49,50]. For example, Tian et al. utilized Lamp2b fused to αv integrin-specific iRGD peptide to target αv integrin-positive breast cancer cells in vitro and in vivo [48]. A dramatic increase in cellular uptake was observed for iRGD-decorated exosomes compared to control exosomes (95.4% vs. 35.0%) [48]. Designer EVs can be used to overcome one of the main challenges faced by traditional therapeutics—crossing the BBB to deliver drugs to the brain. Again, Lamp2b acted as an exosomal pedestal to fuse brain targeting moiety in a study by Alvarez-Erviti et al. They accomplished the targeting of short interfering (si)RNA to the brain in mice by engineering dendritic cells to express Lamp2b conjugated to the neuron-specific rabies viral glycoprotein (RVG) peptide [47].

The designs of protein-targeting constructs that get incorporated into therapeutic EVs can become rather complicated. Ohno et al. used cloned tumour-targeting peptides (EGF and its less mitogenic alternative GE11) into pDisplay vector, which already contained hemagglutinin, myc-tag and platelet-derived growth factor receptor (PDGFR) [51]. Human embryonic kidney cell line 293 (HEK293) cells were then transfected with the pDisplay vector, thus allowing for the incorporation of targeting peptides into exosomes with the PDGFR acting as the carrying pedestal [51]. The targeting modifications with EGF and GE11 peptides increased the uptake of exosomes by breast cancer cells which typically overexpress EGF receptors [51]. At the same time, anti-hemagglutinin and anti-Myc-tag antibodies were used to confirm the expression of EGF and GE11 in exosomes [51]. Johnsen et al. have published a very comprehensive review of EVs as drug delivery vehicles, covering the use of targeting peptides to enhance the precision of EV-based therapeutics [44].

Besides targeting peptides, antibodies and nanobodies have been installed on EV surfaces to improve targeting. In one study, EV-producing cells were transfected with a vector encoding for anti-EGFR nanobodies fused to glycosylphosphatidylinositol (GPI) peptides to target cancer cells [52]. Lipid raft-associated lipids and proteins, including GPI, are naturally enriched on EV membranes, making them ideal for conjugating a targeting ligand [53]. Lactadherin was also used as an anchor for cancer-targeting moiety—single chain variable fragments (scFv) with an affinity to human epidermal growth factor receptor (EGFR) 2 overexpressed in breast cancer cells [54]. This approach was justified as lactadherin associates with the phosphatidylserine enriched in EV membranes.

While peptides and antibodies/nanobodies are the most popular choices for EV target guidance, other more imaginative approaches have also been employed. For example, pseudotyping—a method to change the tropism of viruses by packaging the genetic components of one virus into the envelope proteins of a different virus—has been applied to engineering targeted exosomes [55]. Meyer et al. expressed vesicular stomatitis virus glycoprotein (VSVG), which is frequently used for pseudotyping retroviruses and known for its broad tropism, in HEK293 cells [55]. The resulting VSVG-pseudotyped exosomes were incubated with multiple cell lines, and enhanced uptake by those cells was shown compared to controls [55].

An interesting direction in EV targeting is to use targeting moieties that also have a therapeutic effect. Jiang et al. induced overexpression of tumour necrosis factor (TNF)-related apoptosis-inducing ligand (TRAIL) in the membranes of donor cells which subsequently got incorporated into exosomes loaded with an anti-cancer agent triptolide [56]. In this study, TRAIL acted not only as a targeting ligand for death receptor 5, which is abundant in cancer cells, but also induced apoptosis in cancer cells, thus amplifying the therapeutic effect of triptolide loaded into exosomes [56]. In a different study, MHC-II, a major histocompatibility complex molecule normally only present on professional antigen-presenting cells, was overexpressed in murine melanoma cells [57]. The resulting MHC-II-enriched exosomes not only showed increased targeting towards T cells, but also enhanced the immunological response of the type 1 T helper cell (TH1) against cancer cells [57].

Supplying EVs with targeting properties via genetic modification of parent cells is an effective approach. However, it is not suitable for personalized medicine applications as it is difficult to apply this approach to patients’ own cells. It is also time-consuming, hard to scale for mass production and limiting as only genetically encodable targeting moieties may be used. Moreover, some targeting moieties tend to be expressed incorrectly and are quickly degraded in producer cells, affecting the resulting EVs’ targeting efficiency [58].

### 3.2. Modifying EV Surface for Improved Targeting

Direct chemical modification of EVs allows for a wider selection of targeting ligands and is more suited to personalized medicine applications, as EVs can be isolated from patient’s own body fluids and later decorated with chosen guiding moieties. Further, targeting ligands can be attached to isolated EVs in a very controlled manner which is not achievable with the parental cell modification approach.

Targeting peptides and antibodies/nanobodies are most commonly used to achieve efficient targeting of EVs post-isolation. An interesting approach was suggested by Ye et al., who used a multifunctional peptide to target EVs to glioblastoma cells [59]. The peptide contained a sequence which targeted it to the low-density lipoprotein receptor expressed on the glioblastoma cells, as well as an apoptosis-inducing sequence [59]. This therapeutic/targeting peptide amplified the effectiveness of the chemotherapy agent loaded into EVs and ensured successful delivery over the BBB in a mouse model [59]. Glioma cells were also targeted by Jia et al. via the conjugation of neuropilin-1-targeted peptide to exosomes via click chemistry [60]. The exosomes were loaded with superparamagnetic iron oxide nanoparticles and curcumin to enable both glioma imaging and treatment [60].

Anti-EGFR nanobodies fused to lactadherin, which bind to the phosphatidylserine in EVs post-isolation, have been used to target EGFR-positive tumour cells [61]. Similarly, anti-Her2 single-chain variable fragment fused to lactadherin was conjugated to phosphatidylserine-enriched EVs to target Her2-positive cancer cells [62]. A different approach to surface modification of EVs with nanobodies was adopted by Koojmans et al. who mixed EVs with micelles containing PEG and EGFR nanobodies [63]. This resulted in the emergence of PEGylated EVs targeted to EGFR-positive cancer cells [63].

More out-of-the-box approaches to direct modification of EVs for targeted drug delivery have also been tested. For instance, both DNA and RNA aptamers (single strands of nucleic acid that can bind to specific targets) have been used to direct EVs towards tumours. Cholesterol, naturally present in EV membranes, was used to conjugate AS1411, a DNA aptamer with a high affinity to nucleolin typically overexpressed in breast cancer cells [64]. AS1411-decorated EVs displayed better tumour-targeting abilities and had an added therapeutic effect as AS1411 is known to inhibit tumour activity [64]. An RNA aptamer targeted at prostate-specific membrane antigen (PSMA) has also been loaded into EV membranes to gain directional control [65]. Interestingly, the orientation of this arrow-shaped aptamer could be altered (with either the arrowhead or the arrow tail facing the membrane-anchoring cholesterol) to achieve either better targeting or better cargo loading into EVs [65].

Even actual magnets have been used to direct exosomes towards tumours. Qi at al. decorated transferrin receptors of blood-derived exosomes with magnetic nanoparticles and then guided them towards murine tumours using external magnets [66]. 

A niche and underexplored method for functionalizing EVs with targeting moieties is their fusion with liposomes decorated with targeting peptides or antibodies [67]. For example, Li et al. fused cancer exosomes with liposomes decorated with tumour-targeting peptide cRGD and loaded the resulting hybrid with a chemotherapy drug. The addition of the targeting peptide amplified the natural homing ability of cancer exosomes and resulted in the efficient delivery of the drug in vitro and in vivo [68]. A different study used an exosome-liposome hybrid to treat diabetic peripheral neuropathy. Singh et al. fused exosomes from bone marrow mesenchymal stromal cells with liposomes containing polypyrrole nanoparticles, thus combining stem cell therapy with electrical stimulation to achieve a therapeutic effect. Polypyrrole was selected because it is a conducting polymer and served to target the neurons’ electrical stimulation [69]. Despite some success, there are very few studies using fusion liposomes to achieve the targeting of exosomes.

Overall, the direct EV modification approach allows for a wider selection of targeting ligands and greater freedom in choosing chemical tools for their conjugation. These targeting moieties often have therapeutic and even diagnostic modalities that amplify the effect of loaded cargo drugs. Still, direct modification of EVs is challenging as the reaction conditions must be adapted to preserve EV membranes, prevent aggregation and ensure sufficient density of targeting ligands on EV surfaces.

## 4. Perspectives

EVs have many characteristics that make them an obvious choice as drug delivery vehicles, such as biocompatibility, stability, low immunogenicity, etc. However, the use of EVs as nanocarriers in their natural form remains challenging due to insufficient natural targeting abilities. Multiple attempts have been made to target EVs to specific cells by modifying the producer cells or attaching targeting moieties to EVs post-isolation. While these approaches enhance the interaction of EVs with their target, they do not cancel out EVs’ natural homing abilities and do not prevent off-target effects. Therefore, artificial targeting moieties must be carefully selected to complement EVs’ intrinsic homing abilities. This is a non-trivial task due to the inherent heterogeneity of EV subtypes, which may have different natural targeting ligands, and the absence of standardized testing of EV-based pharmaceuticals. Thus, separating EV subtypes remains a major hurdle for achieving efficient targeting and preventing batch inconsistencies.

While there is an abundance of research on targeting EVs to specific cells, few studies have investigated the ultimate fate of those EVs. EVs are taken up by cells via different routes, including receptor-medicated endocytosis, micropinocytosis and membrane fusion. Furthermore, a large proportion of them may be degraded in the lysosome. More research into EV uptake mechanisms is necessary to ensure they reach the cytosol, and this requires overcoming the current microscopy limitations. Advantage should be taken of EV subtypes that directly fuse with the cellular membrane delivering their contents straight to the cytosol as they do not face the problem of endosomal escape.

Making EV pharmaceuticals commercially and practically viable remains challenging. First, scaling up targeted EV production for pharmaceutical applications is problematic, especially when working with primary cells due to relatively low EV yields. In addition, when modifying EVs post-isolation, the complexity of EV composition may affect the efficiency of conjugation reactions. Creating personalized EV-based medicines is also a highly inefficient process. The time required for the manufacturer to isolate EVs from the patient’s body fluids or primary cells, load them with a drug and perform all the required quality control checks may be considerable, and in the meantime, the patient’s condition may worsen.

Due to the practical difficulties associated with commercializing EVs as drug delivery vehicles, their advantages must be warranted. For many applications, EVs do not offer a substantial advantage over traditional drug delivery vehicles, such as liposomes and polymeric nanoparticles. However, EVs’ unique composition makes them ideally suited to several niche applications. First, EVs’ capacity to deliver therapeutics to hard-to-reach tissues makes them ideal for drug delivery to the brain or the cochlear sensory hair cells. Second, EVs that naturally possess medicinal qualities, such as stem cell-derived EVs, could be used to amplify the therapeutic effect of loaded pharmaceuticals. Third, unlike liposomes and polymeric nanoparticles, EVs are uniquely suited to personalized medicine applications as they could be derived from patients’ own body fluids or cells.

Importantly, care should be taken when selecting producer cells for therapeutic EVs. While there is an abundance of studies on cancer EVs as drug delivery tools due to their natural tumour-targeting abilities, such EVs raise safety concerns as they could potentially transform healthy cells, making them more susceptible to metastasis invasion and resistant to therapy [70,71,72].

Looking to the future, a highly promising but underexplored source of targeted therapeutic EVs is bacterial cells. Bacterial EVs can naturally transfer resistance genes and virulence factors to host cells, but their targeting mechanisms are currently poorly investigated. Bacterial EVs have been successfully loaded with siRNA, gold nanoparticles and antibiotic gentamicin [73]. Importantly, bacterial EVs are naturally immunogenic, making them attractive as potential vaccine components. For example, MeNZB and Bexsero are two bacterial EV-based vaccines against *Neisseria meningitidis* that have been licensed for clinical use [74]. Furthermore, bacterial EVs are perfect natural adjuvants to vaccines as they contain highly immunogenic components (e.g., lipopolysaccharides derived from the outer cellular membrane of Gram-negative bacteria), which stimulate the host’s immune response increasing the effectiveness of exisiting vaccines. For example, Lee et al. used EVs derived from Gram-negative bacteria as adjuvants to enhance the efficacy of the influenza vaccine in a mouse model [75]. The ability of bacterial EVs to modulate immune response could also be used for targeted cancer immunotherapy. For example, systemically administered bacterial EVs accumulated in tumour tissue and triggered the release of antitumour cytokines CXCL10 and interferon-γ, which led to the suppression of tumour growth [76]. However, the immunogenicity of bacterial EVs is a double-edged sword as it can also provoke the undesirable release of inflammatory cytokines, i.e., ‘cytokine storm’, leading to severe complications [77]. Thus, the safety of EVs derived from pathogenic Gram-negative bacteria must be carefully evaluated if they were to be used as therapeutics. EVs from Gram-positive symbiotic and commensal bacteria represent a safer option for therapeutic applications and warrant further investigation.

Further, in addition to delivering therapeutics and vaccines, EV-targeting techniques can also be used for imaging and diagnostics. To achieve this, both genetic engineering of parent cells (to express genes for GFP, RFP, biotin acceptor peptide (BAP), gLuc, etc.) and direct incorporation of dyes (e.g., DiI, PKH67 or PKH26) into the EV membrane have been used [78]. For example, exosomes derived from hypoxic cancer cells and labelled with DiO, a fluorescent lipophilic dye, were preferentially taken up by hypoxic cancer cells compared to cancer exosomes produced under non-hypoxic conditions [40]. Exosomes generated under hypoxic conditions were then labelled with superparamagnetic iron oxide and their accumulation in the hypoxic regions of tumours was successfully monitored in vivo using magnetic particle imaging [40]. Currently, however, most studies aiming at tracking EVs’ distribution and uptake using fluorescent probes utilize non-targeted EVs. A comprehensive review of such studies has been published by Salunkhe et al. [78]. More efforts should be directed at specific targeting of EVs decorated with imaging probes if they are to be used for diagnostic purposes in the clinical setting. 

Ultimately, coordinated effort from EV researchers and start-up owners is necessary to create standardized manufacturing and testing protocols for targeted EV-based therapeutics to ensure their effectiveness, safety and commercial success.

## 5. Conclusions

EVs possess natural targeting abilities that facilitate cell-to-cell communication and deliver molecules to distant sites. EVs can also penetrate brain tissues and the cochlear sensory hair cells, a task that represents a challenge for most synthetic nanocarriers. To translate these findings into clinical applications and utilize EVs as an efficient and reproducible nanocarrier system, the functionalization of EV surfaces with unique targeting ligands is required. This can be achieved by genetically modifying EV producer cells or directly functionalizing the EV surface post-isolation. Both approaches have advantages and should be chosen with the desired application in mind. Notably, the natural targeting properties of EVs should not be disregarded when engineering targeted EVs. Instead, their effect should be amplified further by artificial modifications. The future of targeted EV-based therapeutics lies in combining their natural targeting and medicinal capabilities with designer targeting elements and therapeutic and diagnostic cargo, thus producing a multifunctional biomedical device.

## Figures and Tables

**Figure 1 bioengineering-09-00496-f001:**
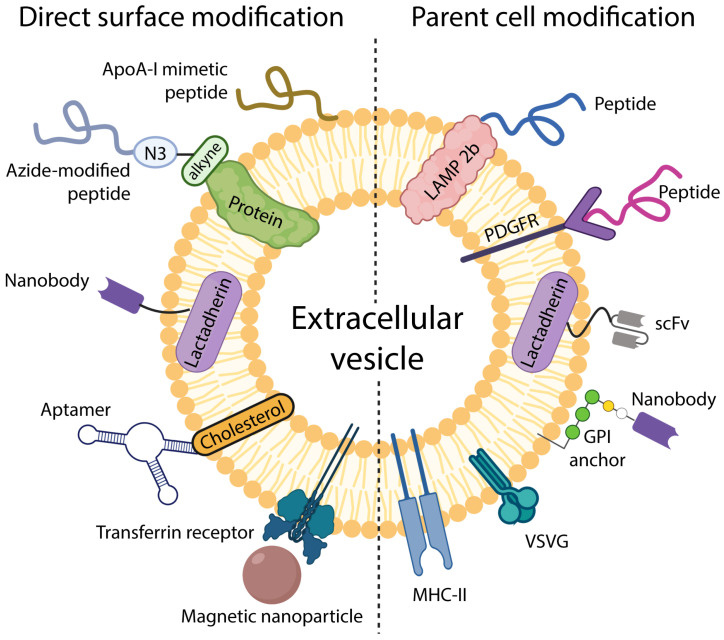
Schematic showing two strategies of EV surface modification. **Left**: methods and moieties used via direct in vitro modification. **Right**: surface modification via parental cell modification.

## Data Availability

Not applicable.

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
