# Peer review of "Targeting Capabilities of Native and Bioengineered Extracellular Vesicles for Drug Delivery"

_bioengineering, 2022, doi:10.3390/bioengineering9100496_

Round 1
Reviewer 1 Report
Dear authors,
Thanks for your contribution on this field. This is an interesting state of art on EVs and how it is possible to modify them for drug delivery. The manuscript is still well written and organized.
All the aspects are almost cover.
Indeed, it will be great if some data can be added on the EV modifications and their used for imaging in human health. This way of modifying them is also related to disease targeting and, in many studies, the next step is to use them for drug delivery. Many examples are already available for the delivery of Doxorubicin in tumor cells where Dox is also used for its “labeling” property.
Also, to complete the manuscript, it will be of interest to add a part on clinical trials, the past one and the ongoing one, with conclusions on obtained results to illustrate the point raised by the authors in the abstract: “Despite ongoing efforts, clinical translation of targeted EV-based drug delivery systems remains a non-trivial task.”
Hoping you can add these points in the manuscript to be accepted for publication.
All the best,
Author Response
Dear Reviewer 1,
Thank you for your valuable comments! We have made significant revisions to the current manuscripts (marked in red) to address your concerns:
- The issue of utilizing targeted EVs for imaging has been addressed in Lines 426-440.
- Past and ongoing trials of EV-based therapeutics are discussed in Lines 89-117.
Many thanks,
Isaac T.S. Li, Liubov Frolova
Reviewer 2 Report
Isaac T. S. Li performed a review on extracellular vesicles (EVs) for drug delivery. The manuscript fell within the scope of Bioengineering. EVs are current research hotspots in the field of biomedical engineering, pharmaceutical development and medical science, and thus I believed that this review could arouse a certain impact. However, some issues should be fixed before the final decision:
(1) With regards to length, the manuscript was a bit short as a Review. Perhaps related content about mechanisms and applications should be expanded. Or, the article type of Mini-review/short-review was suggested.
(2) The word count of Abstract was about 120. In my opinion, the typical word count of an Abstract was 150~250. Please consider to add more important information in the Abstract.
(3) In Line 19, the authors stated that EVs were ‘produced by cells’. This is true, and more detailed information (like what kind of cells?) could be added.
(4) In Line 35, the authors stated that ‘EVs’ capacity to penetrate hard-to-reach tissues is superior to that of synthetic nanocarriers’. Please reconsider this argument. Was it true in all cases, especially for a nanocarrier with sophisticated design? For example, many nanomedicines with high BBB-penetration capacity have been developed.
(5) A schematic illustration could be added to better interpret the self-recognition effects of EVs discussed in Paragraph 2 in Section 2.
(6) Regarding ‘a very detailed review on this topic has been published’ (Line 173), I suggested to give a very brief summary about that review in the current work. Merely stating that a review had been published did not introduce useful information.
(7) For Section 3.2. Modifying EV surface for improved targeting, a possible way to improve targeting was to make a membrane fusion of EVs and other targeting vesicles, e.g. liposomes. Please also conduct a review regarding this aspect.
(8) Maybe, the application of microorganism-derived EVs would become one of the future trends, which could be discussed in the Perspective Section.
(9) Generally, a Conclusion Section was needed for a paper. Probably some content in the Perspective Section could be separated as a Conclusion Section.
(10) Please double-check the format of References.
In summary, a Major Revision was recommended.
Author Response
Dear Reviewer 2,
Thank you for your valuable comments! We have made substantial revisions to address your concerns (new sections are marked in red in the revised manuscript). To address your individual questions:
- The article was indeed meant as a mini-review, and focused on the most recent developments.
- The abstract has been expanded to the appropriate word count.
- The issue was addressed in Lines 22-23.
- The issue was addressed in Lines 39-40.
- Upon careful consideration, we have decided not to add an illustration of the self-recognition effects. Based on our current knowledge, the illustration would be too simplistic for a problem that we have not yet fully understood. Therefore, we do not wish to misguide readers with such figure.
- The issue is addressed in Lines 207-209.
- The topic of EV-liposome fusion hybrids is addressed in Lines 332-344.
- The topic of bacterial EVs is addressed in Lines 402-424.
- A conclusion has been added (Lines 445-458).
- Format of references has been amended.
We would like to thank the reviewer again for the comments, which have helped strengthen this mini-review.
Sincerely,
Isaac T.S. Li, Liubov Frolova
Round 2
Reviewer 2 Report
The authors well addressed the questions raised by me. The current version is acceptable.